# Predictive Factors for a Successful Treatment Outcome in Patients with Different Voiding Dysfunction Subtypes Who Received Urethral Sphincter Botulinum Injection

**DOI:** 10.3390/toxins14120877

**Published:** 2022-12-15

**Authors:** Yao-Lin Kao, Yin-Chien Ou, Kuen-Jer Tsai, Hann-Chorng Kuo

**Affiliations:** 1Department of Urology, National Cheng Kung University Hospital, College of Medicine, National Cheng Kung University, Tainan 704, Taiwan; 2Institute of Clinical Medicine, College of Medicine, National Cheng Kung University, Tainan 704, Taiwan; 3Research Center of Clinical Medicine, National Cheng Kung University Hospital, College of Medicine, National Cheng Kung University, Tainan 704, Taiwan; 4Department of Urology, Hualien Tzu Chi Hospital, Buddhist Tzu Chi Medical Foundation and Tzu Chi University, Hualien 970, Taiwan

**Keywords:** botulinum toxin, urethral, voiding dysfunction, detrusor underactivity, urethral sphincter dysfunction

## Abstract

Voiding dysfunction is a common but bothersome problem in both men and women. Urethral sphincter botulinum toxin A (BoNT-A) injections could serve as an option in refractory cases. This study analyzed the efficacy and outcome predictors of the injections in patients with functional, non-neurogenic voiding dysfunction. Patients who received urethral sphincter BoNT-A injection for refractory voiding dysfunction due to detrusor underactivity (DU) or urethral sphincter dysfunction were retrospectively reviewed. A successful outcome was defined as a marked improvement as reported in the global response assessment. The study evaluated the therapeutic efficacy of urethral sphincter BoNT-A injections and measured the changes in urodynamic parameters after the procedure in the patients. A total of 181 patients including 138 women and 43 men were included. The overall success rate was 64%. A lower success rate was noted in patients with DU compared to those with urethral sphincter dysfunction in both genders. In the multivariable analysis, recurrent urinary tract infection (UTI) and bladder voiding efficiency (BVE) were positive predictors for a successful outcome, while DU was a negative predictor. Urethral sphincter BoNT-A injection is an effective treatment for refractory non-neurogenic voiding dysfunction. Baseline BVE and history of recurrent UTI positively predict a successful outcome. DU is a negative outcome predictor.

## 1. Introduction

Voiding dysfunction is a urological condition characterized by slow or incomplete bladder emptying [1,2]. Being a major component of lower urinary tract symptoms (LUTS) in men, voiding dysfunction is actually not uncommon in women in clinical practice [3]. Detrusor underactivity (DU) and bladder outlet obstruction (BOO) are two fundamental etiologies of voiding dysfunction and both could result from either neurogenic or non-neurogenic origins [4]. The latter could further be subdivided into anatomical obstruction and functional obstruction.

Urodynamic study is often required for the diagnosis of voiding dysfunction. Invasive urodynamic studies such as pressure flow studies or videourodynamic studies (VUDS) could differentiate DU from BOO as causes of voiding dysfunction [5]. VUDS could further undermine the underlying lower urinary tract dysfunction of BOO including urethral stricture, benign prostate obstruction (BPO), high-grade pelvic organ prolapse as anatomical obstruction, primary bladder neck obstruction, or urethral sphincter dysfunction as functional obstructions [6]. The accurate diagnosis and measurement of urodynamic parameters from VUDS may predict and even improve the outcomes of different voiding dysfunctions [7].

Except for simple anatomical obstruction such as BPO or high-grade pelvic organ prolapse, treating entities of voiding dysfunction may be challenging for urologists. Botulinum toxin A (BoNT-A) has been used to treat the neurogenic voiding dysfunction since the late 1980s [8]. Urethral sphincter injection of BoNT-A could decrease urethral resistance and improve voiding efficiency (VE) via chemical sphincterotomy through the blocking of acetylcholine release from presynaptic efferent nerves at the neuromuscular junctions [9]. Benefits of urethral sphincter BoNT-A injections in non-neurogenic voiding dysfunction were also reported afterwards [10,11]. This article aims to explore the effects of urethral sphincter BoNT-A injections in different types of functional, non-neurogenic voiding dysfunction in both genders and search for the predictive factor for treatment outcome.

## 2. Results

There were 181 patients including 138 females and 43 males receiving urethral BoNT-A injections in this study. The mean age at injection was 59.7 ± 21.1 years old in women and 67.3 ± 14.1 years old in men, which was significant younger in the former (*p* = 0.003). Compared to men, women had a higher percentage of recurrent urinary tract infection (43% vs. 5%, *p* < 0.001) and history of receiving transurethral incision or resection of the bladder neck (TUIBN) (50% vs. 19%, *p* < 0.001), but a lower percentage of Parkinson’s disease (1% vs. 14%, *p* = 0.003) and dementia (1% vs. 7%, *p* = 0.042). A total of 56% of men had received transurethral resection of the prostate (TURP). Detailed baseline characteristics and comorbidities stratified by gender are shown in Table 1.

Table 2 shows the baseline and post-injection VUDS parameters and the post-injection GRA in women with different types of voiding dysfunction. There were 61 women with DU and 77 with urethral sphincter dysfunctions in this study. A significantly lower rate of successful outcome was noted in women with DU compared to those with urethral sphincter dysfunction (56% vs. 74%, *p* = 0.024). Except for mild decrease of US, no obvious change of other VUDS parameters was detected in women diagnosed of DU. Increased FSF (103.6 ± 60.5 to 125.3 ± 74.6 mL/s, *p* = 0.034) and decreased Pdet (54.7 ± 36.0 to 45.5 ± 33.9 cmH2O, *p* = 0.034), as well as BOOI (41.8 ± 37.5 to 31.6 ± 35.9, *p* = 0.010), were noted in women with urethral sphincter dysfunction. In the male cohort, there were 43 patients receiving urethral sphincter BoNT-A injection. Among them, 17 men were diagnosed with DU and 26 with urethral sphincter dysfunction (Table 3). A significantly lower rate of successful outcome was noted in men with DU compared to those with urethral sphincter dysfunction (36% vs. 73%, *p* = 0.014). Although there was some dissimilarity in clinical and VUDS parameters, no difference in treatment response rate after urethral sphincter BoNT-A injection was found among different subtypes of urethral sphincter dysfunction (Appendix A Table A1).

Univariable logistic regression analysis for predictors of successful outcome revealed that history of recurrent urinary tract infection (UTI) (OR = 2.37, *p* = 0.024) and VE (OR = 1.02, *p* < 0.001) were positively correlated with the outcome; whereas DU (OR = 0.37, *p* = 0.002), history of hypertension (OR = 0.50, *p* = 0.026) and FS (OR = 1.00, *p* = 0.036) in VUDS correlated with the outcome negatively (Table 4). After adjusting for age and gender, history of recurrent UTI and VE were positive predictors for a successful outcome with odds ratios of 3.82 (95% confidence interval: 1.58–9.22, *p* = 0.003) and 1.02 (95% confidence interval: 1.01–1.03, *p* = 0.004), respectively. On the other hand, DU was a negative predictor with an odds ratio of 0.46 (95% confidence interval: 0.21–0.99, *p* = 0.047) in the multivariable logistic regression analysis.

## 3. Discussion

This study reveals that urethral sphincter BoNT-A injection is an effective treatment option for refractory non-neurogenic functional voiding dysfunction in both genders. The general success (GRA ≧ 2) rate after injection was 64%. Patients with a history of recurrent UTI and favorable baseline VE had better a subjective response after urethral sphincter BoNT-A injections. DU is a significant predictor for poor outcome. These findings suggest that undertaking urodynamic assessment before the procedure is important for predicting treatment outcomes in patients considering urethral sphincter BoNT-A injection due to voiding dysfunction.

The concept of urethral sphincter BoNT-A injection in treating non-neurogenic voiding dysfunction originated from the positive experience of its usage in treating patients with detrusor sphincter dyssynergia [12]. It was assumed that the improvement of voiding dysfunction is related to lowering of the urethral resistance through chemical sphincterotomy which was induced by blocking the presynaptic release of acetylcholine in the neuromuscular junction of the urethral sphincter after injection [13]. Previous studies have reported about 60–70% overall response rate in non-neurogenic voiding function after the urethral sphincter BoNT-A injections [14,15]. With a greater sample size, our studies demonstrated a similarly successful result in such patients. Notably, high proportion of our patients had history of TUI-BN or TURP. In our practice, we performed TUI-BN in female patients who presented with insufficient bladder neck opening during voiding in the videourodyamic studies prior to urethral sphincteric BoNT-A injection. This treatment sequence could exclude the patients whose voiding dysfunction was attributed to anatomical or functional bladder neck dysfunction. A similar rationale was also applied to the male patients; TURP or TUI-BN were performed first if obstruction in the prostate urethra or bladder neck was suspected. In short, urethral sphincteric BoNT-A was injected in patients with refractory voiding dysfunction due to DU or urethral sphincter dysfunction. In the logistic regression analysis, history of TUI-BN or TURP did not pose significant adverse effects to the outcome of urethral sphincteric BoNT-A injection.

DU and urethral sphincter dysfunction are the two major etiologies of non-neurogenic voiding dysfunction. However, studies comparing the treatment efficacy between the two are lacking and the study subjects were often mixed with those with neurogenic voiding dysfunction [15,16]. DU was reported to be one of the causes of treatment failure after urethral sphincter BoNT-A injections [16]. In this study, both women and men had a significantly lower rate of treatment success in patients diagnosed with DU compared to those with urethral sphincter dysfunction. For DU patients, the major effect of urethral sphincter BoNT-A injection is to release BOO by lowering the urethral resistance while the impaired bladder contractility persisted despite the treatment. This could explain the inferior outcome in these patients. After adjusting for the possible confounding factors including gender difference and age, DU remains a predictive factor for poor treatment response in this study.

It is reasonable that the therapeutic efficacy of urethral BoNT-A might be affected by the severity of baseline pathophysiology of voiding dysfunction. Patients with history of urethral catheterization due to severe emptying failure in idiopathic or neurogenic etiology had been reported to respond poorly to the treatment compared to others [17]. As an index of bladder emptying ability, VE before the treatment might work as an outcome predictor as well. In fact, we found that the baseline VE was positively correlated with the successful outcome in both univariate and multivariate analyses in this study. The best cutoff value for baseline VE were 23% and 4 % for females and males respectively according to the Youden’s index in the receiver operating characteristic (ROC) curve. Therefore, patients with DU and poor VE diagnosed in pre-operative urodynamic studies should be adequately informed of the risk of inferior treatment responses.

Aside from the therapeutic effect for voiding function, urethral sphincter BoNT-A injection might also be beneficial for recurrent UTI, a common bothersome nightmare resulting from incomplete urine emptying [18]. Urethral sphincter BoNT-A injection had been reported to achieve a 50% reduction of UTI in spinal cord injury patients with detrusor sphincter dyssynergia [19]. Urethral sphincter BoNT-A injection also decreased UTI in neurologically normal patients with functional voiding dysfunction [20]. The benefit of UTI prevention might explain the finding of higher subjective response rates reported in our patients who had a history of recurrent UTI. As a result, urethral sphincter BoNT-A injection might be considered in those who suffered from refractory voiding dysfunction concomitant with recurrent UTI.

This study provides the treatment response rate of urethral sphincter BoNT-A injection and its predictive factors in patients suffering from functional, non-neurogenic voiding dysfunction with considerable subject numbers as well as complete and detailed urodynamic study before and after the treatment. However, there are still some limitations. First, since the majority of male voiding dysfunction is caused by anatomical obstruction, the number of men in our study is relatively small which makes it difficult to undertake further subgroup analysis. Second, the diagnoses of DU and urethral sphincter dysfunction were based on the image and pressure flow parameters from VUDS which might be somewhat subjective. Nevertheless, it is the most common way to differentiate the cause of voiding dysfunction in clinical practice. Third, the retrospective nature of this study made it difficult to avoid all possible biases during analysis despite our adjusting for the significant variables statistically. A prospective trial with specific inclusion criteria and pre-defined sub-group analysis is required to confirm the results of our study.

## 4. Conclusions

Urethral sphincter BoNT-A injection is effective in treating refractory functional non-neurogenic voiding dysfunction in both genders. The overall successful rate was 64%. Baseline VE and history of recurrent UTI positively correlate with a successful outcome. DU is a predictive factor for a poor treatment outcome.

## 5. Materials and Methods

The study was initiated following approval by the Institutional Review Board of the author’ hospital (IRB 105-151-B). From January 2010 to November 2019, patients who received urethral sphincter BoNT-A injection due to refractory functional, non-neurogenic voiding dysfunction were retrospectively reviewed. All patients available for baseline and follow-up VUDS data were included. Patients with anatomical urethral conditions including uncorrected BPO and high-grade pelvic organ prolapse, history of lower urinary tract reconstruction, urethral stenosis and urethral tumor were excluded. Patients with uncorrected bladder neck dysfunction, neurogenic abnormality related detrusor sphincter dyssynergia, cauda equina syndrome or peripheral neuropathy were also excluded [21]. The voiding dysfunction among patients with cerebral vascular accident, Parkinson’s disease or dementia with subtle neurological clinical manifestation were not considered neurogenic since the lower urinary tract symptoms manifest in these diseases were predominantly detrusor overactivity with or without incontinence [22]. The bladder neck dysfunction was corrected first if the VUDS revealed a narrow bladder neck during the voiding phase in patients with voiding dysfunction. Because the patients still have difficulty in urination after TUI-BN or TUR-P, they were recommended to receive urethral sphincter BoNT-A injection for the urethral sphincter dysfunction.

VUDS performed in accordance with the International Continence Society (ICS) recommendation [23] were utilized for baseline urinary function assessment of every patient with refractory voiding dysfunction before the urethral sphincter BoNT-A injection. The cause of voiding dysfunction was determined by VUDS and electromyography (EMG) as DU or urethral sphincter dysfunction. DU was defined as having a bladder contractility index ≦ 100 in men, and maximal detrusor pressure (Pdet) < 10 cm H_2_O with maximum flow rate (Qmax) < 10 mL/s and post-void residual (PVR) > 150 mL in women. The external urethral sphincter dysfunction was subclassified into dysfunctional voiding (DV) or poor relaxation of the external sphincter (PRES) according to the features of VUDS and EMG. DV was diagnosed as the stasis of contrast at the level of urethral sphincter presenting with the typical feature of a “spinning top” urethra during the voiding phase of VUDS with increased external urethral sphincter EMG activity at the same time [24]. PRES was defined as the narrowing of the distal urethra without the presentation of a “spinning top” urethra during the voiding phase of VUDS without the concomitant relaxation of the external urethral sphincter EMG activity [25].

Other parameters of VUDS included first sensation of filling (FSF), full sensation (FS), urge sensation (US), compliance in the storage phase and Pdet, Qmax, BOO index (BOOI), PVR, cystometric bladder capacity (CBC) and VE in the voiding phase. The bladder contractility index was calculated as Pdet + 5× Qmax and BOOI was calculated as Pdet—2 × Qmax [26]. CBC was calculated by voided volume plus PVR in the VUDS. VE was defined as the voided volume divided by the CBC in the VUDS. Major comorbidities including diabetes mellitus, hypertension, chronic kidney disease, chronic obstructive lung disease, coronary artery disease, and neurogenic disease beyond the sacral spinal cord–brainstem pontine micturition center pathways, as well as history of recurrent urinary tract infection, transurethral resection of the prostate (TUR-P) or transurethral incision or resection of the bladder neck (TUI-BN) were collected from medical records.

All patients received 100 units onabotulinumtoxinA (BOTOX, Allergan, Irvine, CA, USA) external urethral sphincter injections in the operation room under light intravenous general anesthesia [27]. The location of the external urethral sphincter was identified by direct visualization under cystoscopy in both men and women. In male patients, urethral sphincter injections were performed transurethrally using a 23-gauge needle (22 Fr, Richard Wolf, Knittlingen, Germany) with 4–8 injections circumferentially distributed in the external urethral sphincter at a depth of 0.5 cm along the longitudinal direction of the urethral lumen. Female patients, on the other hand, received urethral sphincter injections periurethrally using 27-gauge 1 mL syringe needles with 4–8 injections circumferentially into the external urethral sphincter at a depth of 1.5 cm along the longitudinal direction of the urethral lumen. A detailed description of the urethral sphincter injection technique was reported in our previous review [28]. Treatment outcomes of urethral sphincter BoNT-A injections were assessed at around 3 months after the procedure since the average therapeutic duration was reported at around 6 months [9]. Subjective outcomes were measured by global response assessment (GRA) as excellent (+3), markedly improved (+2), mildly improved (+1) or no change (0), according to the patients’ perception of the voiding condition after the BoNT-A injections. Patients with an excellent outcome can get rid of the catheter, and patients with marked improvements still need CIC occasionally. A successful outcome was defined as GRA equal to or greater than 2. Objective outcomes were also assessed by VUDS follow-up after the injections.

All analyses were performed through SAS Statistics for Windows, Version 9.4, Cary, NC, USA: SAS Inc. Two-sided *p*-values less than 0.05 were considered significant. The continuous variables of baseline demographics were expressed as the mean ± standard deviation whereas the categorical ones were expressed as number (percentage). Differences between gender of the above variables were examined with independent t-test in continuous variables and chi-square test in the categorial ones. We applied Fisher’s exact test in circumstances when more than 20% of the expected frequencies were less than five. Changes in post-treatment variables in each gender were examined with the paired samples t-test and McNemar test for continuous and categorical variables, respectively. The distribution of the GRA grades after injections between DU and urethral sphincter dysfunction was examined with the chi-square test. Univariate logistic regression analysis was performed to find out the predictive factors for a successful treatment outcome. Variables demonstrating significant differences in the univariable analysis, including age and gender, were further evaluated in the multivariable model.

## Figures and Tables

**Table 1 toxins-14-00877-t001:** Baseline characteristics and comorbidities stratified by gender.

	Female (*n* = 138)		Male (*n* = 43)		*p* Value
	Mean ± SD or No. (%)		Mean ± SD or No. (%)		
Age	59.7	±21.1	67.3	±14.0	0.003
Diagnosis					
Detrusor underactivity	61	(44)	17	(40)	
Urethral sphincter dysfunction *	77	(56)	26	(60)	0.589
Diabetes mellitus	36	(26)	11	(26)	0.947
Hypertension	65	(47)	18	(42)	0.547
CAD	13	(9)	1	(2)	0.193
CKD	3	(2)	1	(2)	1.000
COPD			1	(2)	0.238
Parkinson disease	2	(1)	6	(14)	0.003
CVA	19	(14)	7	(16)	0.682
Dementia	1	(1)	3	(7)	0.042
Recurrent UTI	47	(34)	2	(5)	<0.001
History of TURP			24	(56)	<0.001
History of TUI-BN	69	(50)	8	(19)	<0.001

CKD: Chronic kidney disease; COPD: Chronic obstructive pulmonary disease; CVA: Cerebrovascular accident; UTI: Urinary tract infection; TURP: Transurethral Resection of Prostate; TUI-BN: Transurethral Incision or Resection of the Bladder Neck. * Urethral sphincter dysfunction including dysfunctional voiding and poor relaxation of urethral sphincter.

**Table 2 toxins-14-00877-t002:** Baseline and post-injection urodynamic parameters and the post-injection global response assessment in female patients with different types of voiding dysfunction.

Female (*n* = 138)	Detrusor Underactivity					Urethral Sphincter Dysfunction *				
	Before Urethral Botox Injection (*n* = 61)		After Urethral Botox Injection (*n* = 61)		*p* Value	Before Urethral Botox Injection (*n* = 77)		After Urethral Botox Injection (*n* = 77)		*p* Value
	Mean ± SD or No. (%)		Mean ± SD or No. (%)			Mean ± SD or No. (%)		Mean ± SD or No. (%)		
VUDS parameters										
FSF (mL)	177.3	±76.6	158.6	78.2	0.152	103.6	±60.5	125.3	±74.6	0.034
FS (mL)	250.1	±86.4	230.0	104.6	0.142	170.3	±78.8	193.2	±98.1	0.059
US (mL)	295.7	±106.5	266.3	109.6	0.042	204.5	±98.3	219.9	±109.5	0.242
Compliance (mL/cm H_2_O)	64.3	±80.1	58.0	50.4	0.585	46.3	±62.2	56.3	±64.6	0.258
DO	7	(11)	6	(10)	0.655	51	(66)	42	(55)	0.083
Pdet(cm H_2_O)	6.4	±7.9	7.2	±11.1	0.573	54.7	±36.0	45.5	±33.9	0.009
Qmax (mL/s)	4.0	±7.1	4.9	±6.5	0.460	6.5	±4.9	6.9	±5.5	0.430
BOOI	−1.4	±16.4	−2.5	±13.7	0.623	41.8	±37.5	31.6	±35.9	0.010
VV (mL)	89.3	±131.5	101.0	±139.5	0.596	124.2	±102.5	138.8	±124.0	0.337
PVR (mL)	315.4	±210.5	313.3	±220.5	0.952	187.4	±142.8	200.9	±159.0	0.480
BVE (%)	21.5	±30.9	26.1	±35.0	0.402	42.4	±31.9	45.1	±35.8	0.540
Global Response Assessment										
Excellent			5	(8)				20	(26)	
Markedly improved			29	(48)				37	(48)	
Mildly improved			8	(13)				5	(6)	
No change			18	(30)				14	(18)	
Missing			1	(2)				1	(1)	0.024 ^b^
Successful outcome ^a^			34	(56)				57	(74)	0.024 ^b^

BOOI: bladder outlet obstruction index; BVE: bladder voiding efficiency; DO: detrusor overactivity; DU: detrusor underactivity; FS: full sensation; FSF: first sensation of filling; Pdet: maximal detrusor pressure; PVR: post-void residual volume; Qmax: maximal uroflow rate; SD: standard deviation; US: urge sensation; VV: voided volume; VUDS: videourodynamic study. ^a^ Successful outcome was defined as a global response assessment greater than mildly improved (score ≧ 2). ^b^ Difference between detrusor underactivity and urethral sphincter dysfunction. * Urethral sphincter dysfunction including dysfunctional voiding and poor relaxation of urethral sphincter.

**Table 3 toxins-14-00877-t003:** Baseline and post-injection urodynamic parameters and the post-injection global response assessment in male patients with different types of voiding dysfunction.

Male (*n* = 43)	Detrusor Underactivity					Urethral Sphincter Dysfunction *				
	Before Urethral Botox Injection (*n* = 17)		After Urethral Botox Injection (*n* = 17)		*p* Value	Before Urethral Botox Injection (*n* = 26)		After Urethral Botox Injection (*n* = 26)		*p* Value
	Mean ± SD or No. (%)		Mean ± SD or No. (%)			Mean ± SD or No. (%)		Mean ± SD or No. (%)		
VUDS parameters										
FSF (mL)	181.4	±105.9	172.2	±99.1	0.727	146.2	±89.6	150.8	±68.3	0.818
FS (mL)	265.9	±153.0	275.3	±156.2	0.828	248.8	±116.6	251.6	±134.0	0.919
US (mL)	320.6	±157.3	323.4	±164.4	0.947	283.9	±124.3	281.5	±142.1	0.934
Compliance (mL/cmH_2_O)	73.5	±141.5	41.1	±38.0	0.359	44.5	±44.6	56.4	±50.7	0.424
DO	4	(24)	6	(35)	0.157	9	(35)	13	(50)	0.103
Pdet(cm H_2_O)	10.5	±11.0	18.0	±24.2	0.285	24.2	±17.5	23.5	±17.3	0.791
Qmax (mL/s)	1.8	±2.6	2.5	±2.6	0.282	5.5	±5.3	6.8	±6.0	0.271
BOOI	6.9	±9.7	12.9	±24.2	0.336	13.3	±16.5	10.0	±15.1	0.405
VV (mL)	24.5	±40.4	38.9	±50.6	0.258	138.5	±137.0	138.6	±150.8	0.997
PVR (mL)	375.3	±162.0	393.8	±185.4	0.712	237.4	±176.5	231.1	±191.8	0.831
BVE (%)	8.2	±12.5	11.4	±13.7	0.348	40.1	±35.2	44.1	±39.0	0.552
Global Response Assessment										
Excellent			3	(18)				7	(27)	
Markedly improved			3	(18)				12	(46)	
Mildly improved			1	(6)				1	(4)	
No change			10	(59)				6	(23)	0.094 ^b^
Successful outcome ^a^			6	(36)				19	(73)	0.0141 ^b^

BOOI: bladder outlet obstruction index; BVE: bladder voiding efficiency; DO: detrusor overactivity; DU: detrusor underactivity; FS: full sensation; FSF: first sensation of filling; Pdet: maximal detrusor pressure; PVR: post-void residual volume; Qmax: maximal uroflow rate; SD: standard deviation; US: urge sensation; VV: voided volume; VUDS: videourodynamic study. ^a^ Successful outcome was defined as a global response assessment greater than mildly improved (score ≧ 2). ^b^ Difference between detrusor underactivity and urethral sphincter dysfunction. * Urethral sphincter dysfunction including dysfunctional voiding and poor relaxation of urethral sphincter.

**Table 4 toxins-14-00877-t004:** Logistic regression analysis for predictors of successful outcomes after urethral sphincteric botulinum toxin A injection.

	Univariate Analysis					Multivariate Analysis		
	OR	95% CI		*p* Value	OR	95% CI		*p* Value
Age	0.98	0.97	1.00	0.062	1.00	0.98	1.02	0.728
Gender	0.72	0.36	1.45	0.353	0.88	0.38	2.04	0.769
DU	0.37	0.20	0.70	0.002	0.46	0.21	0.99	0.047
Comorbidities								
DM	0.61	0.31	1.19	0.147				
HTN	0.50	0.27	0.92	0.026	0.53	0.25	1.13	0.099
CAD	2.17	0.58	8.06	0.250				
CKD	1.70	0.17	16.66	0.649				
PD	0.93	0.22	4.02	0.923				
CVA	1.07	0.45	2.56	0.882				
Dementia	0.55	0.08	4.02	0.558				
Recurrent UTI	2.39	1.12	5.09	0.024	3.82	1.58	9.22	0.003
TURP history	0.62	0.26	1.48	0.280				
TUIBN history	0.80	0.43	1.47	0.462				
Baseline VUDS parameters								
FSF (mL)	1.00	0.99	1.00	0.159				
FS (mL)	1.00	0.99	1.00	0.036	1.00	1.00	1.00	0.559
US (mL)	1.00	1.00	1.00	0.164				
Compliance (mL/cm H_2_O)	1.00	1.00	1.00	0.500				
DO	1.59	0.84	3.00	0.155				
BVE (%)	1.02	1.01	1.03	<0.001	1.02	1.01	1.03	0.004
BOOI	1.01	1.00	1.02	0.327				

BOOI: bladder outlet obstruction index; BVE: bladder voiding efficiency; CAD: Coronary artery disease; CI: confidence interval; CKD: Chronic kidney disease; COPD: Chronic obstructive pulmonary disease; CVA: Cerebrovascular accident; DM: diabetes mellitus; DO: detrusor overactivity; DU: detrusor underactivity; FS: full sensation; FSF: first sensation of filling; HTN: hypertension; PD: Parkinson’s disease; PVR: post-void residual volume; Qmax: maximal uroflow rate; UTI: Urinary tract infection; TURP: Transurethral Resection of Prostate; TUI-BN: Transurethral Incision or Resection of the Bladder Neck; US: urge sensation; VUDS: videourodynamic study.

## Data Availability

Data are available on request to the corresponding author.

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
