# Peer review of "Predictive Factors for a Successful Treatment Outcome in Patients with Different Voiding Dysfunction Subtypes Who Received Urethral Sphincter Botulinum Injection"

_toxins, 2022, doi:10.3390/toxins14120877_

Round 1

Reviewer 1 Report

1) The tables are very overloaded (!) and confusing. It would make more sense to concentrate primarily on significant values ​​and say goodbye to rather uninteresting, non-significant values. 2) How do you explain a TUIBN of 50% in women (Table 1), that's a very high rate of this kind of operation in women! Was an anatomical narrowing the cause before a functional dysfunction was then treated with BTX? 3) How do you explain a DO in patients with a DU in 11% of women and 24% of men?? 4) Did I miss the frequencies of UTIs before/after BTX? I only find the odds ratios. That indeed would be interesting, how this UTI rate was reduced. 5) How long did the therapy last? Because the duration of sphincter injetction is usually quite short lasting.

Reviewer 2 Report

Dear authors, congratulations on this work on such a difficult subject, from the diagnosis to the treatment!

Some considerations are written and some are in my text commentaries.

Please, discuss the lack of urethra EMG during the UD/VUDS and how it would be important to this study and these conditions

Please discuss the inclusion of 9 neurogenic patients (Parkinson + VCA) in the male population, since you present this study as “non-neurogenic”.

Please discuss the response to urethral BTX in the male PdetQmax (table 3).

Most of the urodynamic positive results were seen in the subgroup analysis of table 4. Maybe the efficacy of the treatment is highly correlated with the diagnosis more than with other parameters.

Of the cohort were any patient(s) that were using indwelling catheter/CISC and were able to be free of them?

Reviewer 3 Report

The findings of the study are useful. I would like to provide several suggestions.

(1)  In male patients, both internal and external urethral sphincter were injected via the transurethral route? How many sites were injected for each of the two sphincters? In female patients, only external urethral sphincter was injected? If so, the two genders received different injection targets and such a difference may influence the outcomes.

Based on lines 196-199, dysfunctional voiding (DV) was diagnosed as the stasis of contrast at the level of urethral sphincter and presenting with the typical feature of “spinning top” urethra [23] during voiding phase. Poor relaxation of the external sphincter (PRES) was defined as narrowing of the distal urethra without the presentation of “spinning top” urethra during the voiding phase of VUDS [24]. Then, the contraction of both internal and external urethral sphincters leads to dysfunctional voiding (DV) and thus both sphincters should be treated (whereas only the external sphincter should be treated for PRES)?

(2)  I suggest that the authors introduce the injection technique with more details, e.g., depth and direction (angle) of the injection, with one or two figures. What aspects should be noted during injection? Do variations of injection techniques influence the outcome?

(3)  For men and women, how to identify the location of urethral sphincter during injection?

(4)  How long was the duration between the baseline and the follow-up VUDS data?

(5)  In tables 2 and 3, post-void residual volume (PVR) was at the level of 200-300ml after injection, and slightly higher than those before injection. Why didn’t PVR decrease when Botox injection improved relaxation of urethral sphincters?

Line 60-62, “TUI-BN” and “TUR-P”. But in Table 1 and its legend, “TUIBN” and “TURP”. These are inconsistent.

Table 2. The legend did not provide the full names of many abbreviations.

Line 146, “Receiver Operating Characteristic Curve (ROC)” should be “Receiver Operating Characteristic Curve (ROC) curve”.

Round 2

Reviewer 2 Report

Thank you for your kind response. 

I'm happy with it.